# Real-time observation of two distinctive non-thermalized hot electron dynamics at MXene/molecule interfaces

Qi Zhang[1,2], Wei Li[3,4], Ruixuan Zhao[1], Peizhe Tang [5], Jie Zhao[2], Guorong Wu [2], Xin Chen [3,4], Mingjun Hu[5], Kaijun Yuan [2,6,7] ✉, Jiebo Li[1] ✉ & Xueming Yang [2,6,8]

The photoinduced non-thermalized hot electrons at an interface play a pivotal role in determining plasmonic driven chemical events. However, understanding non-thermalized electron dynamics, which precedes electron thermalization (~125 fs), remains a grand challenge. Herein, we simultaneously captured the dynamics of both molecules and non-thermalized electrons in the MXene/molecule complexes by femtosecond time-resolved spectroscopy. The real-time observation allows for distinguishing non-thermalized and thermalized electron responses. Differing from the thermalized electron/heat transfer, our results reveal two non-thermalized electron dynamical pathways: (i) the non-thermalized electrons directly transfer to attached molecules at an interface within 50 fs; (ii) the non-thermalized electrons scatter at the interface within 125 fs, inducing adsorbed molecules heating. These two distinctive pathways are dependent on the irradiating wavelength and the energy difference between MXene and adsorbed molecules. This research sheds light on the fundamental mechanism and opens opportunities in photocatalysis and interfacial heat transfer theory.

The utilization of light to drive plasmon-induced chemical reactions has emerged as a highly promising method for initiating chemical transformations on metal nanostructures[1–6]. Understanding when and how energy is dissipated from metal to attached molecules at an interface, containing both electron and heat transfer processes, is critical for reshaping the chemical reaction landscape, accelerating reaction rate, and controlling the selectivity[7–14]. In bare metal, the optical excitation of a plasmon with energy (hv) exceeding its Fermi level ($E_f$) will immediately create energetic electrons (hot electrons). These initially non-thermalized hot electrons can persist till the end of

electron-electron scattering in tens to hundreds of femtoseconds, reaching electron thermalization with a defined electronic temperature following the Fermi-Dirac distribution[9,15–18]. Subsequently, the thermalized electron undergoes a cooling process to heat lattice (phonon) within a few picoseconds through electron-phonon coupling[19–21]. In a metal/molecule complex, the molecules can serve as acceptors to harvest hot electron or hot lattice energy. Conventional wisdom acknowledges, gained from noble metal nanomaterials, that the interfacial electron transfer takes place after electron thermalization (i.e., the thermalized electron transfer (TET), shown

[1]Institute of Medical Photonics, Beijing Advanced Innovation Center for Biomedical Engineering, School of Biological Science and Medical Engineering, Beihang University, Beijing 100191, P.R. China. [2]State Key Laboratory of Molecular Reaction Dynamics and Dalian Coherent Light Source, Dalian Institute of Chemical Physics, Chinese Academy of Sciences, 457 Zhongshan Road, Dalian 116023, P.R. China. [3]Suzhou Laboratory, Suzhou 215123 Jiangsu, China. [4]GuSu Laboratory of Materials, Suzhou 215123 Jiangsu, China. [5]School of Materials Science and Engineering, Beihang University, Beijing 100191, P.R. China. [6]Hefei National Laboratory, Hefei 230088, China. [7]University of Chinese Academy of Sciences, Beijing 100049, China. [8]Department of Chemistry and Center for Advanced Light Source Research, College of Science, Southern University of Science and Technology, Shenzhen 518055, China. ✉ e-mail: kjyuan@dicp.ac.cn; jiebo39@buaa.edu.cn

in Supplementary Fig. 1a–c)[4,9,10,22,23] but before the electron cooling process in weak coupling interfaces. While, the interfacial heat transfer takes place after electron-phonon coupling, followed by the dissipation of thermalized lattice energy into the adjacent surface molecules through interfacial vibration-vibration coupling (namely, the phonon-mediated heat transfer (PMHT), displayed in Supplementary Fig. 1a, b, d, e)[24–26] in serval picoseconds. To date, all reported initial ultrafast processes (-100 fs) at the interface involve charge transfer[27–30], while ultrafast interfacial heat transfer in tens to hundreds of femtoseconds seems to be impossible.

The novel atomically thin nanosheet, featuring weak van der Waals interaction in the out-of-plane direction and representing fascinating electrical, optical properties[31,32], opens opportunities in photocatalysis and beyond[33,34]. The two-dimensional transition metal carbide (MXene) is a new generation of plasmonic materials. Differing from semiconductor nanosheets, it exhibits metallicity and plasmon features from visible to infrared band[35]. It is also different from noble metal nanostructures in terms of geometry, relatively inert surface, and narrowband plasmonic absorption. The metallic MXene advancements in atomically thin layers, the complex surface electronic properties[36–41], and broadband optical absorption characteristics[42], which may provide a unique reaction platform for exploring electron and heat transfer at MXene/molecule interfaces. However, to our best knowledge, there are limited reports about the non-thermalized hot electron dynamics at this interface. To gain a comprehensive understanding of electron/heat transfer at interfaces, it is necessary to acquire transient spectral responses and dynamics for both non-thermalized electrons and molecules till the end of electron-electron scattering.

In this work, we employed femtosecond pump-probe spectroscopy to demonstrate the energy migration from MXene to molecules. Specifically, we investigated this phenomenon with MXene nanosheets and a series of adsorbed molecules at the interface. The experiment achieves real-time distinction between non-thermalized and thermalized electron responses at the interface during the extremely fast electron thermalization process[43,44]. Our results verify two direct non-thermalized electron dynamical pathways: for the MXene/Methylene blue (MB) complex, the non-thermalized electrons exhibit a direct transfer from MXene to the attached MB within a remarkably short timeframe of 50 fs (Fig. 1 pathway I), bypassing electron-electron scattering. In contrast, for MXene/Rhodamine 6 G (R6G) complex,

the energy loss of non-thermalized electrons occurs via the electron scattering at MXene/R6G interface, resulting in heat transfer to attached R6G molecules within the initial 125 fs (Fig. 1 pathway II). Remarkably, pathway II represents a new heat transfer channel with the rate being an order of magnitude faster than the conventional one. These two pathways shed light on the direct interaction between the non-thermalized electrons and attached molecules. Furthermore, we demonstrate that these energy flow channels can be modulated by altering either the irradiating wavelength or surface molecules. This research provides crucial insights into the intricate energy dynamics at the nanoscale interface and offers a foundation for potential applications in controlling and manipulating chemical reactions.

## Results

### Steady-state and transient optical characterizations of MXene/molecules film

To investigate the direct interaction between non-thermalized electrons and molecules, we selected metallic MXene ($Ti_3C_2T_x$) nanosheets as the non-thermalized electron donor. The $T_x$ represents surface terminated groups of MXene, such as -O, -F, -OH. The choice of MXene was driven by its ultrathin structure, which minimizes energy loss for non-thermalized electrons in MXene. We selected surface molecules with cationic properties and planar configurations due to their enhanced absorption characteristics when contacting the MXene surface[45,46]. Detailed information on MXene synthesis[42] and the preparation of MXene/molecule complexes can be found in the Methods. The characterization data (Supplementary Fig. 2) reveal that the MXene used in this study possesses a thickness of approximately 2 nm and a diameter ranging from 300 to 1000 nm. The transmission electron microscope of MXene/molecules has also been shown in Supplementary Fig. 3. To acquire the dynamics of non-thermalized electrons and molecules, the femtosecond time-resolved pump-probe spectroscopy was employed (as detailed in Methods). As shown in Fig. 2a, a single wavelength pulse serves as the pump to excite the MXene, generating excited electrons. Subsequently, a supercontinuous white light is employed as the probe to monitor signals from both the molecules and the MXene, capturing the response of interfacial excited electrons. In Fig. 2b, the steady-state absorption spectrum of MXene exhibits a broad absorption band spanning from 650 nm to 900 nm, with a peak at approximately 750 nm

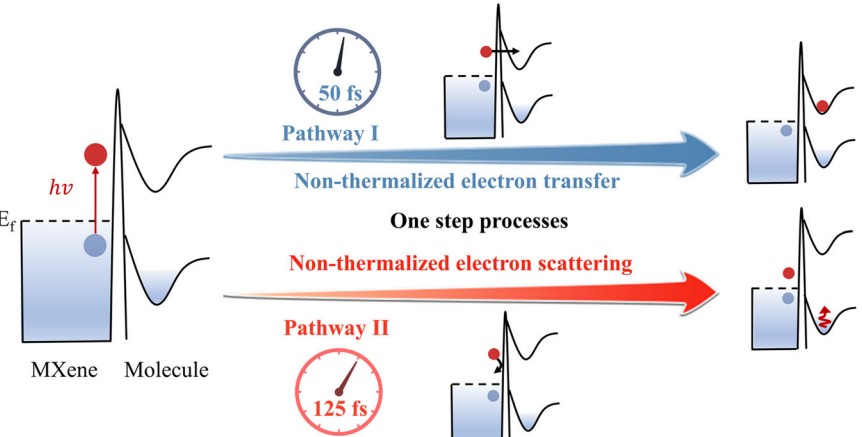

**Fig. 1 | Schematic illustration presenting two different energy transfer dissipation pathways from the non-thermalized electron to the molecule at interface after optical excitation.** The non-thermalized electron transfer (pathway I), where the non-thermalized hot electrons directly transfer to molecules without experiencing electron-electron scattering; the non-thermalized electron-induced heat transfer (pathway II), where the non-thermalized hot electrons scatter

at MXene/molecules interface, with the lost energy in the scattering process instantaneously heating the molecules. The red and blue balls represent an electron and a hole, respectively. The region above the Fermi surface (dashed line) corresponds to the MXene conduction band, and the curves correspond to molecular potential energy surfaces. The connecting line between both elements corresponds to an energy barrier.

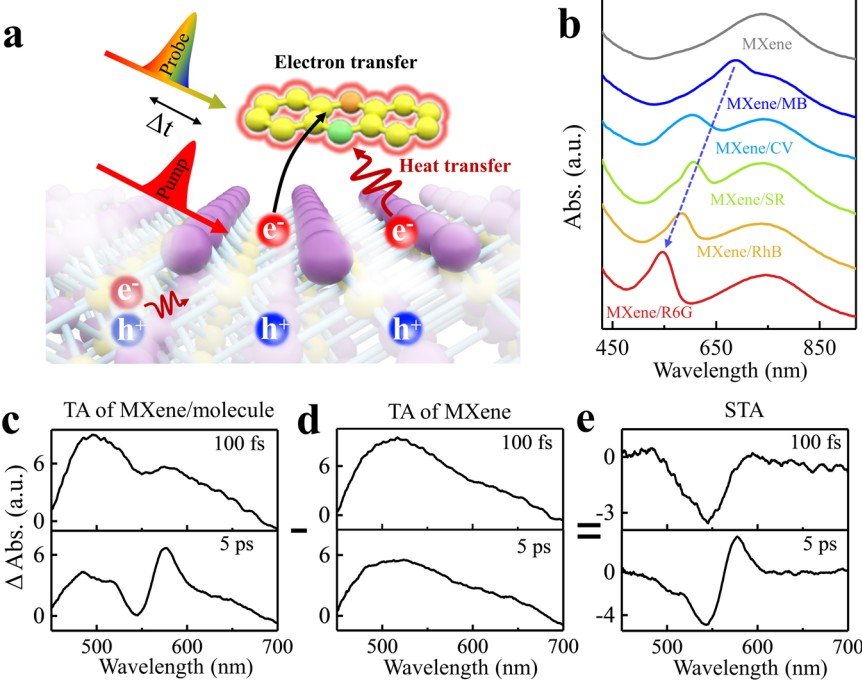

**Fig. 2 | Steady-state and transient absorption spectra of the designed MXene/molecule systems. a** The experiment for an ultrafast pump-probe approach uses a wavelength-switchable laser as the pump to generate nonthermal electron-hole pairs in MXene. Then a super-continuous white-light source is used to probe the molecular response at different time delay. Some electrons on MXene's surface may interact with molecules, leading to both electron and heat transfer. The remaining electrons decay in MXene, resulting in the lattice heating. The purple spheres represent transition metal atoms in MXene, and the yellow spheres represent carbon atoms. The terminations on MXene surface are omitted in the simplified schematic diagram. **b** Steady-state absorption spectra of the MXene/molecule film. The a.u. in the y-axis label represents arbitrary unit. **c, d** The transient absorption spectra of MXene/molecule and MXene, respectively. **e** The subtracted transient absorption spectra (STA), clearly showing molecular ground state bleach (GSB) at 100 fs and 5 ps.

(corresponding to 1.58 eV) attributed to the localized transversal surface plasmon (SP)[35,47]. After the molecules including R6G, Rhodamine B (RhB), Sulfarhodamine 101 (SR), Crystal violet (CV), and MB adsorbing onto the MXene surface, the MXene/molecule complexes display an additional discrete peak at approximately 680 nm (1.82 eV), 602 nm (2.06 eV), 602 nm (2.06 eV), 583 nm (2.12 eV) and 545 nm (2.27 eV) respectively, which arises from the molecular absorption. It is noted that, with the surface molecules changing from MB to R6G, the spectral overlap between the molecular absorption and the SP band decreases gradually.

The pump-probe transient absorption (TA) spectra of the MXene/molecule complex (as illustrated in Fig. 2c and Supplementary Fig. 4a) reveal a broad photoinduced absorption (PA) peak at around 510 nm, originating from MXene, along with a distinct peak contributed by the molecule when pumping at 800 nm (1.55 eV). The TA spectra of MXene are presented in Fig. 2d and Supplementary Fig. 4b, which are consistent with previous observations[42,47]. The broad positive band 500–550 nm originates inter-band transition according to the calculated band structures of MXene (Supplementary Fig. 5a). The electron-electron scattering or heat makes an increased number of electrons near the Fermi level, resulting in photoinduced enhanced inter-band transitions as shown in Fig. 2d. The photo-induced absorption peak of ~510 nm for MXene in TA spectra comes from the overlap between inter-band absorption and plasmonic band (Supplementary Fig. 5b). By subtracting the TA spectra of MXene (Fig. 2d) from that of the MXene/molecule complex (Fig. 2c), the net molecular spectra (the negative signal) at different waiting times can be obtained, which are referred to as subtracted TA spectra (STA) (Fig. 2e). The combined analysis of TA and STA allows us to elucidate the time-resolved energy flow dynamics from MXene to molecules, providing valuable insights into the interaction between these components.

## Simultaneous capturing of the transient spectra of both MXene and molecules

To elucidate the dynamics of non-thermalized electrons and molecules, we initiated our investigation by examining the TA spectra of the MXene/MB film system, which serves as a model where the SP energies match the energy level of molecular transition upon excitation at 800 nm. It's worth noting that the small area ratio of molecules to the MXene surface (approximately 12%, as detailed in Supplementary Note 1) was deliberately maintained to prevent molecular aggregation. Upon exciting the SP of MXene, several pathways may facilitate the transfer of non-thermalized electrons to molecules, like PMHT, TET, or plasmon induced direct electron transfer/molecular excitation within 10 fs[6,28,48–50]. These processes occur in different timescales and exhibit distinct dynamic features. The TA spectra of MXene/MB in Fig. 3a represent a composite signal originating from both MXene and MB. This enables us to simultaneously capture two physical processes: electron-electron scattering in MXene and the interaction between non-thermalized electrons and MB. The spectral range of 500-550 nm can be exclusively attributed to MXene in the TA spectra. The observed signal growth in this segment, spanning a waiting time of 0-125 fs, is an indication of the electron-electron scattering process (EES) occurring in MXene. The analysis of the growing dynamics at 510 nm provides an EES time of ~120 fs (Supplementary Fig. 6), which is consistent with the decay time of non-thermalized electrons[27,42,51]. Figure 3b displays the subtracted TA spectra of MB (STA-MB), representing the response of MB molecular signal induced by the excited electrons in MXene. The results indicate that the STA-MB signal appears at ~0 fs and reaches the maximum value at ~50 fs. This means the molecular response signal growth process (0-50 fs) is significantly faster than the EES process (0-125 fs), suggesting the electron transfer from MXene to MB should be faster than the EES process with excitation at 800 nm. Such electron

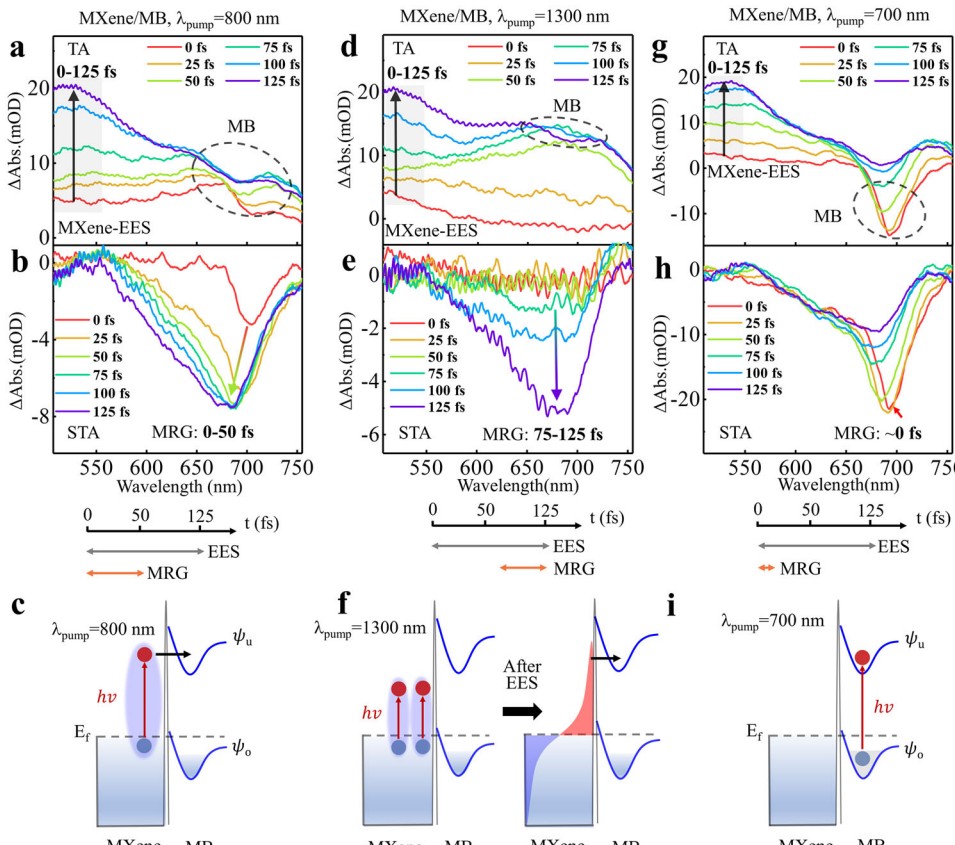

**Fig. 3 | Pump wavelength-dependent molecules response growth (MRG) at the MXene/MB interface. a**, **b** The TA and STA-MB spectra with the pumping wavelength at 800 nm. **c** The schematic diagram presents the non-thermalized electron transfer pathway. **d**, **e** The TA and STA-MB spectra with the pumping wavelength at 1300 nm. **f** The schematic diagram presents the thermalized electron transfer pathway. **g**, **h** Present the TA and STA-MB spectra with a pumping wavelength at 700 nm. **i** The schematic diagram illustrating direct molecular excitation. EES represents the electron-electron scattering process for MXene. In Fig. **a**, **d**, and **g** the gray area represents MXene's signal, and the area outlined by the black dashed line represents MB's signal. In the TA spectra, black arrows indicate MXene's electron-electron scattering process with time delay, while colored arrows indicate the MRG process. The lines representing the time scales belong to the elements of the Fig. **b**, **e**, and **h**. The EES stands for electron-electron scattering. The $\psi_o$ and $\psi_u$ represent occupied states and unoccupied states, respectively, in MB.

transfer differs from the previously reported TET process[22,23], in which the electron transfer occurs following the EES process. This direct non-thermalized electron transfer (NET) from MXene to MB, which bypasses the electron-electron scattering, can be named pathway I, and the schematic shown in Fig. 3c.

We have performed a control experiment by utilizing a lower photon energy of 1300 nm (0.95 eV) to excite another surface plasmon mode[35]. The absorption spectrum of MXene, extending to the near-infrared range (Supplementary Fig. 7), demonstrates that photons at 1300 nm can indeed be absorbed by MXene. The instrumental response functions (IRF) for pump pulses at both 800 nm and 1300 nm were found to be identical (Supplementary Fig. 8). Comparing with the results at 800 nm pumping, the TA and STA spectra with excitation at 1300 nm (Fig. 3d, e) reveal that there is no MB signal observed within the initial 50 fs. The STA signal appears at -75 fs and reaches its maximum at -125 fs. It seems that the sequence of the EES process precedes the electron transfer, which is the case of TET, as depicted in Fig. 3f. This experiment suggests that the electron transfer pathways can be effectively altered by changing the irradiating wavelength. We also conducted a control experiment with excitation at 700 nm, in which the photon energy is above the energy level of MB molecular transition (Fig. 3g, h). The STA-MB's signal reaches its maximum at -0 fs, representing the direct excitation of MB molecules (Fig. 3i) or resonant energy transfer.

We then studied the STA spectra of the MXene/R6G film system, which serves as a model where there is a large energy mismatch between the SP and the energy level of molecular transition. The TA

and STA spectra within the initial 125 fs for MXene/R6G with excitation at 800 nm are presented in Fig. 4a, b. The EES process also can be seen in the spectrum range around 500 nm, which spans a waiting time of 0–125 fs. The STA-R6G spectrum shows the negative signal appears at 0 fs and increases gradually until 125 fs. Furthermore, the responses of both TA and STA spectra with excitation at 1300 nm (Fig. 4c, d) are similar to those at 800 nm. These are quite different from those observed in MXene/MB. The STA-R6G's signal reaches its maximum at -0 fs with excitation at 520 nm, representing the direct excitation of R6G molecules (Fig. 4e, f).

**Probing the molecular dynamics feature in MXene/molecules complexes**

To abstract more information, we acquired the dynamics from the STA spectra of the MXene/MB and MXene/R6G film systems. Figure 5a displays the dynamics of the STA-MB with probing at the negative peak position (-675 nm). The dynamics with excitation at both 700 and 800 nm exhibit a similar decay process. The growth process is much faster at 700 nm due to direct molecular excitation. The decay process (0.1-0.6 ps) with a fitting time of -100 fs in STA-MB at 800 nm pumping is in line with MXene's electron cooling (fitting time constant of $110 \pm 15$ fs) resulting from electron-phonon coupling (as shown in Supplementary Fig. 9). These results suggest that the transferred electrons in the excited state of MB return to MXene and decay in MXene. This phenomenon has been seen in previous studies[9,27], as illustrated in Fig. 5b. Interestingly, there remains a non-decay

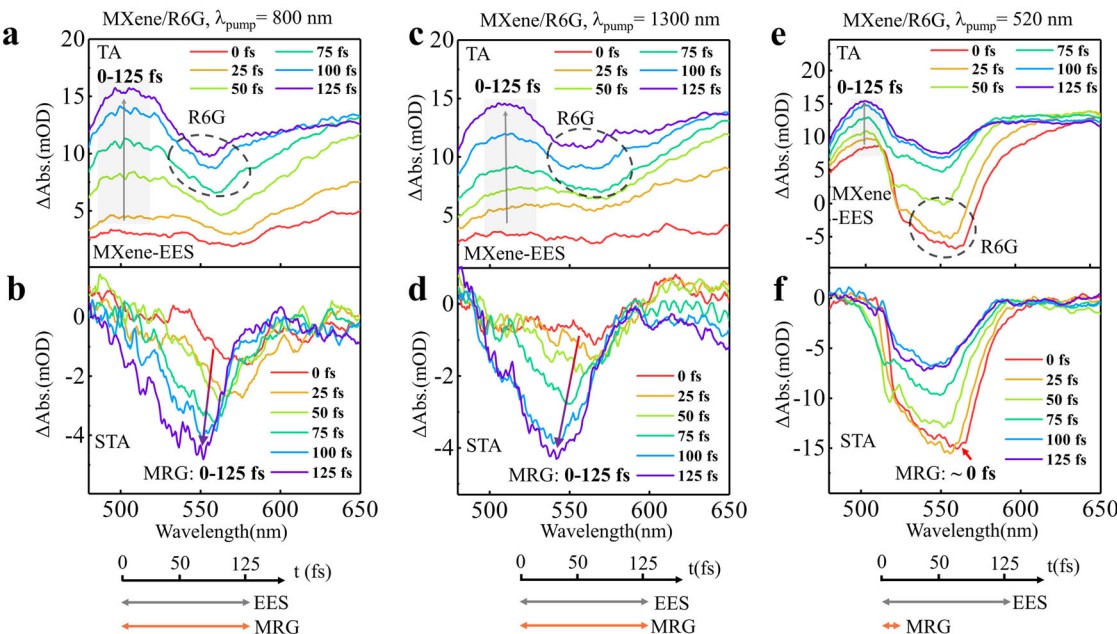

**Fig. 4 | Pump wavelength-dependent molecules response growth (MRG) at the MXene/R6G interface.** The TA and STA-MB spectra with the pumping wavelength at **a**, **b** 800 nm **c**, **d** 1300 nm **e**, **f** 520 nm, respectively.

component (13%) in the STA-MB dynamics. This phenomenon is different from previous studies in noble metal/semiconductor and semiconductor/molecule systems, where complete decay was observed[27–29].

The source of the non-decay component is not immediately clear. We then analyzed the dynamics of the MXene/R6G system. The steady-state absorption spectra for isolated MXene and R6G were shown in Supplementary Fig. 10. In this case, there is a large energy gap between the absorption peak of R6G (~2.27 eV) and MXene's plasmon (~1.65 eV). Obviously, by using the pump wavelength of 800 nm (1.55 eV), the MXene's SP is excited rather than R6G. Furthermore, the electron transfer from MXene to R6G is expected to be more difficult owing to the large energy difference. Figure 5c shows the STA-R6G dynamics probed at the negative peak position (545 nm). It is surprising that after experiencing a rapid growth process within approximately 125 fs, the STA dynamics also exhibit a non-decay behavior during the period from 0.1 to 0.6 ps under 800 nm excitation. Such response couldn't arise from electron transfer, since the transferred electron would usually decay through electron transfer back to the metal, as reported in previous studies[27–29] and illustrated in Fig. 5b.

To verify the absence of electron transfer in MXene/R6G under 800 nm excitation, we conducted a control TA 0experiment for the MXene/R6G system under excitation at 520 nm. The photon energy of 520 nm is sufficient to directly excite R6G molecules. The STA-R6G dynamics at 520 nm (Fig. 5c) show a rapid decay process (0.1-0.6 ps) with a time constant of $115 \pm 12$ fs. The decay time constant is consistent with the electron cooling time in MXene ($110 \pm 10$ fs), but inconsistent with the decay time constant ($1.2 \pm 0.1$ ps) observed in isolated R6G (Supplementary Fig. 11a−c). These results suggest that the excited electrons in R6G dominantly transfer to MXene and decay in MXene (Fig. 5b), rather than directly decay in R6G to generate heat (Supplementary Fig. 12). In other words, if an electron enters the R6G's unoccupied orbital, either through excitation or electron transfer, the electron would decay rapidly within 0.1-0.6 ps in MXene/R6G system. Therefore, the non-decay dynamics observed in STA-R6G with 800 nm excitation strongly suggest that electron transfer is not a dominant process in MXene/R6G. It is noted that the non-decay feature cannot be explained by the phonon-mediated heat transfer process, which

typically occurs in several picoseconds. In addition, the dynamics of TA-MXene (probed at 545 nm) with 800 nm excitation are also shown in Fig. 5c. Interestingly, the timescale of the growing dynamics of STA-R6G is consistent with the rising dynamics of TA-MXene, which are ascribed as MXene's EES process. This similarity suggests that the STA-R6G under 800 nm excitation originates from non-thermalized electron scattering. Such non-thermalized electron scattering at the MXene surface would induce the adsorbed molecule heating. To verify this, we further measured the temperature differential spectrum of MXene/R6G (reflecting the change in absorption spectrum due to temperature, Supplementary Fig. 13a and Note 2). The similarity between the STA-R6G spectrum under 800 nm excitation and the temperature differential spectrum with $\Delta T = 35$ K is observed (Supplementary Fig. 13a, b). The results confirm that the non-thermalized electron scattering directly heats R6G, as schematically shown in Fig. 5d. We term this process non-thermalized electron-induced direct heat transfer (NEIHT). This pathway (~125 fs) is much faster than the well-known PMHT process in the picosecond time scale. It is noted that we demonstrate two distinctive pathways (non-thermalized electron and heat transfer) in MXene/MB and MXene/R6G systems. To further clarify this, we conducted polarization-dependent TA experiments for MXene/MB and MXene/R6G after excitation at 800 nm. The polarized TA and STA spectra within 100 fs after excitation at 800 nm in MXene/MB and MXene/R6G are presented in Supplementary Figs. 14,15. Remarkably, the polarized TA and STA spectra (Supplementary Fig. 14) at 0 fs exhibit significant anisotropy in MXene/MB, while MXene/R6G (Supplementary Fig. 15) at 0 fs displays isotropic behavior. As the time evolves to 100 fs (Supplementary Fig. 16a−c), the anisotropy in STA-MB decays, while STA-R6G remains isotropic. The different anisotropy behavior confirms the non-thermalized electron and heat transfer pathways in MXene/MB and MXene/R6G.

Especially, the NEIHT pathway can be modified by changing the photon energy. We conducted pump-wavelength-dependent TA experiments for MXene/MB and MXene/R6G. The STA-R6G dynamics under excitation at different pump wavelengths (650, 700, 800, and 1300 nm) are illustrated in Supplementary Fig. 17. We can assess the contribution of the NEIHT pathway based on the population of the non-decay component (NDC). For the MXene/R6G and MXene/MB system, the population of NDC is summarized in Fig. 5e, obtained

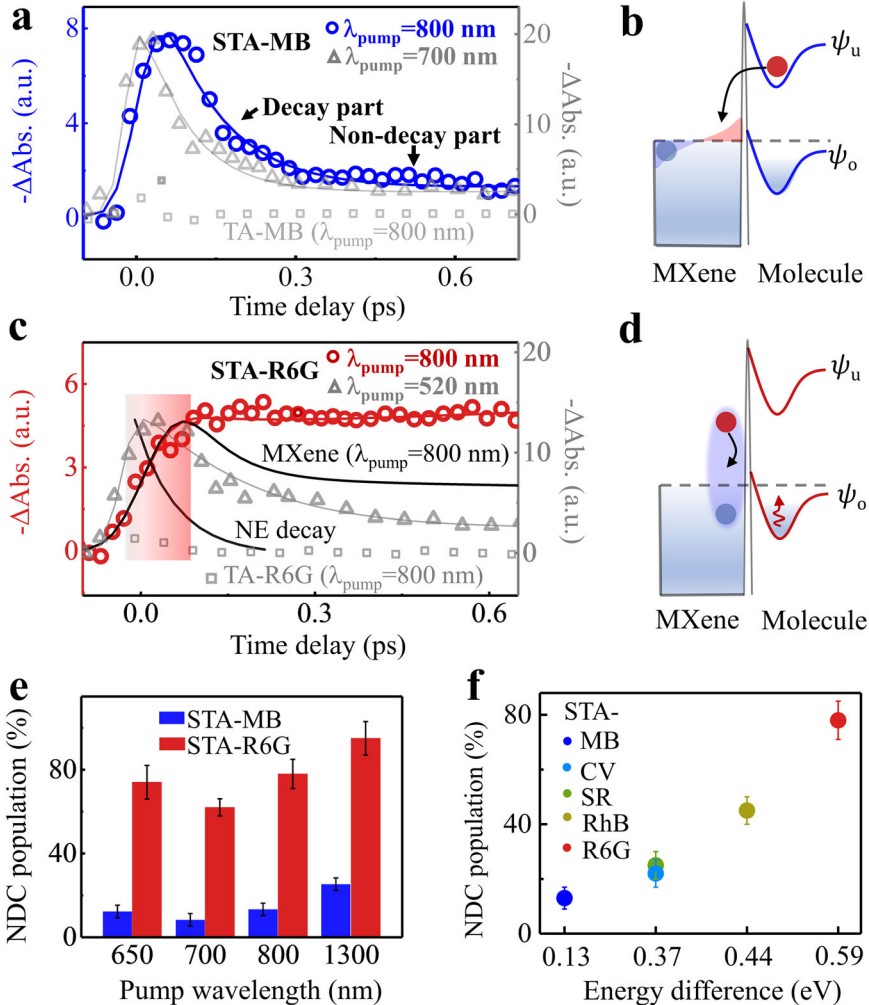

**Fig. 5 | Two distinctive pathways at MXene/molecule interfaces. a** The STA-MB dynamics probed at the peak position after excitation at 800 nm(circles) and 700 nm(triangles). The squares represent experimental TA dynamics at the peak position for isolated MB. The lines represent fitting results. **b** The schematic diagram illustrating the electron back-transfer to MXene. **c** The STA-R6G dynamics probed at the peak position after excitation with 800 nm (circles) and 520 nm(triangles). The squares represent experimental TA dynamics at the peak position for isolated R6G. The lines represent fitting results. The highlighted red portion underscores the similarity in the growth processes of STA-R6G and MXene under excitation at 800 nm. The NE decay represents the simulated non-

thermalized electron decay dynamics, which are obtained by utilizing an exponential decay with the time constant acquired from the electron-electron scattering of MXene. **d** The schematic diagram presents non-thermalized electron scattering (black arrow) and the red arrow represents heat. **e** Non-decay component (NDC) population modulated by the pump wavelength for MXene/MB and MXene/R6G. **f** NDC population modulated by the energy difference between MXene and molecules at the pumping wavelength of 800 nm. The NDC populations are obtained by fitting the decay curves with single exponential. The error bars represent the fitting errors.

through fitting (see Supplementary Note 3) the STA dynamics data in Supplementary Figs. 17 and 18. These results reveal that the NDC is a major component (60-92%) in MXene/R6G and a minor component (5-15%) in MXene/MB with various pump wavelengths. Generally, the population of the NDC increases with increasing pump wavelength, except at 700 nm. To exclude the pump pulses directly exciting R6G or heating the $CaF_2$ substrate, control experiments involving TA spectra of an isolated R6G film with pump beams at 650, 700, and 800 nm were conducted (Fig. 5c and Supplementary Fig. 19). These experiments show no observable signals. The NEIHT pathway can also be modified by changing the energy differences between MXene and adsorbed molecules. The additional STA experiments with MXene/RhB, MXene/SR, and MXene/CV were performed under excitation at 800 nm (Supplementary Figs. 20–22). The populations of the NDC in different systems are summarized in Fig. 5f. The results reveal that the population of NDC increases as the energy difference between MXene and adsorbed molecules increases. It is rational that a large energy

mismatch will suppress the electron transfer, and thus promote the heat transfer.

## DFT supporting the non-thermalized electron and heat transfer

To gain a deeper understanding of the two interfacial energy transfer mechanisms, we performed theoretical calculations involving interfacial adsorption energies, charge density differences, and species-distinguishable density of states (DOS). Specifically, we analyzed the MXene/R6G and MXene/MB systems. In Fig. 6a, the charge density difference reveals the redistribution of charges after the combination of R6G with MXene. This analysis shows that R6G is strongly adsorbed on MXene's surface with an energy of −4.6 eV and a short interfacial distance of 1.8 Å. In the entire MXene/R6G system, the DOS for R6G, presented in Fig. 6b, includes the highest occupied and lowest unoccupied molecular orbital states (HOMO and LUMO), labeled as $\psi_o$ and $\psi_u$, respectively. The DOS for MXene displays two states below and above the Fermi energy, labeled as $\psi_{om}$ and $\psi_{um}$, respectively. When

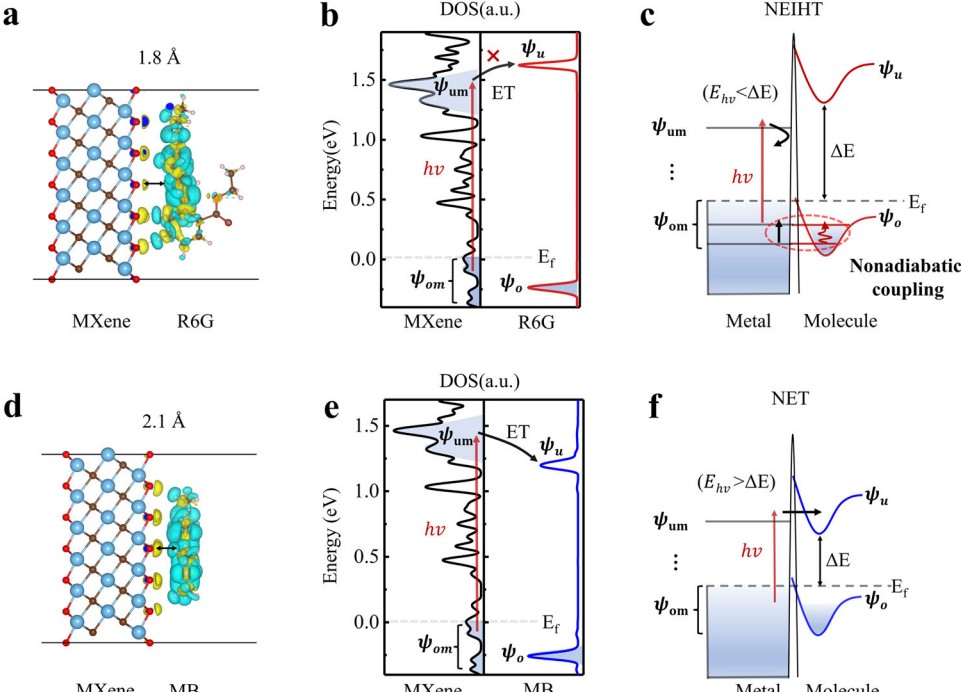

**Fig. 6 | First-principle calculations and proposed the non-thermalized electron and heat transfer mechanisms. a** Calculated charge density differences of MXene/R6G. The yellow and cyan areas represent a gain and loss of electrons, respectively. **b** Partial density of states of entire MXene/R6G. The $\psi_{om}$ and $\psi_{um}$ represent occupied states and unoccupied states, respectively, in MXene. **c** The schematic diagram presents non-thermalized electron interface scattering induced molecular heating (NEITH) by non-adiabatic coupling with nuclear motion state in molecular $\psi_o$ electronic ground state, and electronic states in MXene's $\psi_{om}$. **d** Calculated charge density differences of MXene/MB. **e** Partial density of states of entire MXene/MB. **f** The schematic diagram presents the non-thermalized electron transfer (NET) mechanism.

absorbing one 800 nm photon, an electron in the $\psi_{om}$ state of MXene is excited to the $\psi_{um}$ state. However, the energy level of the $\psi_{um}$ state (1.46 eV) is lower than the interfacial potential barrier (1.62 eV) which prevents the excited electron injecting into R6G's $\psi_u$ state. This inhibits the non-thermalized electron transfer pathway, and the non-thermalized electrons would transfer their energy to other electrons located in the occupied states near the Fermi level via electron-electron scattering. When the molecular nuclear motion resonates with the energy levels of the metal's electronic transitions, as shown in the region marked with a red dashed circle in Fig. 6c, the metal's electronic state couples with the molecular vibrational state at the interface. This forms the interfacial electron-nuclear coupling between the electronic state in $\psi_{om}$ and the nuclear motion state in $\psi_o$, a phenomenon known as nonadiabatic coupling[52–56]. Previous study results showed the energy transfer from molecular vibrations to the continuum of e-h pair excitations via the nonadiabatic coupling[53,54]. While our results exhibit that the energy flow is in the opposite direction, which the generated non-thermalized eleectron in the $\psi_{um}$ state transfers energy via direct scattering with the interfacial electron-nuclear coupling, leading to the molecule heating, as shown in Fig. 6c. Furthermore, our proposed physical picture, based on experimental results, also aligns with the pioneer's theoretical predictions[57], in which the energy of non-thermalized hot electrons could be released into molecular nuclear vibrations. In contrast, for MXene/MB, the calculated results (Fig. 6d) show the interfacial charge redistribution, strong adsorption energy (−3.7 eV), and a short interfacial distance (2.1 Å). The DOS of the entire MXene/MB (Fig. 6e), demonstrates that when MXene is excited at 800 nm, an electron in the $\psi_{om}$ state jumps to the $\psi_{um}$ state (1.46 eV), and the lower interface potential barrier (1.20 eV) allows the excited electron inject into MB's $\psi_u$ state. Consequently, the **NET** mechanism dominates in MXene/MB (Fig. 6f).

## Discussion

Our findings unravel the presence of two distinct electron/heat transfer pathways from MXene to surface molecules. These results highlight two critical factors that determine the direction of energy migration in these strong interaction interfaces: the energy gap (ΔE) between MXene and the attached molecules and the illuminating photon energy ($E_{h\nu}$). For the electron transfer to occur, the excited electron should possess sufficient energy ($E_{h\nu} > \Delta E$) to overcome the interfacial energy barrier and inject into the molecule from MXene. Conversely, when $E_{h\nu} < \Delta E$, the electron transfer is a minor process, and much of the non-thermalized electrons scatter at the interface, leading to molecule heating. Hence, manipulating the values of $E_{h\nu}$ and ΔE can modify the population of these two pathways.

In addition, the strength of interfacial interaction between the MXene and the attached molecules is also crucial for both processes. Our DFT calculations (Fig. 6a and d) show the high adsorption energies (≥3.7 eV) and small distances (≤2.1 Å) between MXene and molecules, which indicate effective interfacial spatial wavefunction overlap. This overlap facilitates strong interfacial electron-electron and electron-nucleus interactions. Previous studies[41] also reported that the MXene exhibits relatively strong interaction with molecules, including electrostatic, polymerization, and strong binding. The relatively strong interaction leads to NET/NEIHT mechanisms occurring at MXene/molecule interfaces. Furthermore, the structures of ultrathin layers also contribute to the occurrence of the NET/NEIHT pathways. In contrast, noble metals like silver and gold have fully occupied d orbitals, thus providing fewer electrons when interacting with molecules, leading to relatively weaker adsorptions with the adsorption energy being less than 1 eV, and the distance between the molecule and the metal being greater than 2.9 Å[58]. Besides, the ligand molecules on the surface of noble metal nanostructures further reduce the

interaction with molecules. The weak interaction leads to the conventional electron/heat transfer pathways in most of noble metal/molecule systems.

In summary, we captured the transient spectral response of non-thermalized electrons and molecules in the MXene/molecule complexes. Two distinctive non-thermalized electron/heat transfer pathways, namely non-thermalized electron transfer (Fig. 1, pathway I) and non-thermalized electron induced heat transfer (Fig. 1, pathway II), have been verified for the first time, to the best of our knowledge. The pathway I differs from the well-accepted TET[9,10,22] route. Notably, the pathway II represents a novel heat transfer route, in which the non-thermalized electron-induced heat transfer occurs within 125 fs. This pathway originates from non-thermalized electron scattering at the interface and exhibits a significantly faster time scale compared to other heat transfer pathways[25,26,59–61]. Additionally, nonthermalized hot holes may also inject into molecules or transfer energy to molecular vibrations. In any case, our systematic studies reveal that the energy flow pathway from non-thermalized carrier to attached molecules dependents on the pumping wavelengths and the nature of the binding molecules. These findings hold the potential to advance plasmonic applications and further develop existing theories of interfacial heat transfer.

## Methods

### Synthesis of $Ti_3C_2T_x$ MXene nanosheet
The synthesis of $Ti_3C_2T_x$ MXene nanosheets (Shandong Xiyan New Material Technology Co., Ltd., 5 mg/mL) followed a previously established procedure[42]. In detail, 1.5 g of LiF was mixed with 20 mL of 9 M hydrochloric acid in a polytetrafluoroethylene beaker. The solution was stirred for 5 minutes. Subsequently, 1.0 g of $Ti_3AlC_2$ powder was slowly added to the etching solvent over 10 minutes. The mixture was stirred at 35 °C for 24 h. Following the reaction, the product was subjected to centrifugation and washed with 1 M hydrochloric acid to remove excess LiF. The product was then diluted with deionized water and subjected to centrifugation (3500 rpm for 5 minutes) several times until the pH of the supernatant solution reached 6–7. Afterward, it was dispersed in deionized water and sonicated for 3 hours under a flow of argon gas. Finally, the solution was centrifuged at 1200 g for 60 minutes to obtain a colloidal solution of few- or single-layer $Ti_3C_2T_x$ flakes.

### Preparation of MXene/molecules complexes
The stable colloidal MXene solution was divided into two parts. The first part of the solution was used to determine the concentration of MXene. This involved suction filtration and drying to obtain the dry MXene powder, which was then weighed. The concentration of the MXene solution was subsequently calculated based on the weight of MXene and the volume of the solution. In the next step, the other part of the MXene aqueous solution with a predetermined concentration was diluted to achieve a concentration of approximately ~0.2 mg mL$^{-1}$. Separate aqueous solutions of molecules were prepared with a concentration of $2.0 \times 10^{-5}$ M by dispersing solid molecules into distilled water. Subsequently, equal volumes of the two aqueous solutions (MXene and each molecule) were mixed, shaken, and sonicated for 1 minute. The resulting mixed solution had concentrations of approximately ~0.1 mg mL$^{-1}$ for MXene and $1.0 \times 10^{-5}$ M for the molecules. Then, the MXene/molecule film sample was obtained by dripping the mixed solution onto a 1-millimeter-thick $CaF_2$ window and drying for several hours.

### The femtosecond transient absorption spectroscopy
The femtosecond ultrafast laser beams used in these time-resolved experiments were derived from the Ti: sapphire laser system (coherent Vitesse and Coherent Legend Elite He+USP-1K-III, ~800 nm, 35 fs, 7 mJ/pulse, and 1 kHz repetition rate). The part of the fundamental laser was

used to inject an optical parametric amplifier (OPA, light conversion: Topas+UV/vis) system, in which the tunable wavelength pump beams (240–2400 nm) can be generated as pump pulse. A weak energy part of the fundamental pulse (1–2 μJ/pulse) was used to generate a super-continuum white light (450-750 nm) by focusing on a sapphire plate as a probe pulse. Both pump and probe beams were focused and overlapped on the samples. The time delays were controlled by a motorized delay line (ALS10045-S-M-10-MT-LT45AS-CM, Aerotech) to change the optical path between the pump and probe beams. The step length was set up to 20/25 fs during 0–1.5 ps. A synchronized chopper was used to chop the pump pulse to 500 Hz. A neutral-density filter was used to adjust the fluence of the pump pulse. A λ/2 waveplate was used to change the direction of polarization of the pump beam. The probe beam enters a fiber spectrometer (AvaSpec-ULS2048CLEVO, Avantes) to record the signal of the change in transient photoinduced transmission. A polarizer located at the front of the fiber spectrometer was used to eliminate pump scattering. Owing to electron and heat transfer at MXene/molecule interfaces, it leads to the difference in TA signal sizes between the MXene/molecules and MXene at non-overlapped regions with molecule signal under identical pump fluence. To obtain accurate STA spectra, the sizes of MXene's TA spectra were corrected based on the TA size of MXene/molecules at 510 nm or non-overlap region by multiplying a coefficient. For example, the result is show in Supplementary Fig. 23.

### DFT calculation methods
The first-principles DFT calculations were performed using the projector-augmented wave method and the Perdew-Burke-Ernzerhof exchange-correlation functional, as implemented in the Vienna Ab initio Simulation Package[62–64]. To describe the effects of the long-range van der Waals interactions, a semiclassical dispersion correction scheme (DFT-D3) was employed[65]. The plane-wave energy cutoff was set to 500 eV, and the convergence threshold for the iteration in a self-consistent field was set at $10^{-5}$ eV. All the atoms could relax until the forces exerted on each atom were less than 0.02 eV/Å during the structural optimization. The 2D Brillouin zone was sampled using a $19 \times 19 \times 1$ mesh per unit cell. The equilibrium lattice constant for $Ti_3C_2O_2$ was 3.015 Å. The molecular adsorption systems were simulated by a $6 \times 6 \times 1$ supercell. To avoid any interaction between the periodic images, calculations were performed with ~30 Å vacuum space. The charge transfer between molecules and the $Ti_3C_2O_2$ layer was calculated using the Bader charge method and VASPKIT script[66,67]. The DFT calculations focus on oxygen-terminated MXene because both the energy dispersive spectroscopy (Supplementary Fig. 3b) and a previous study[68] indicate that the oxygen termination dominates on the surface of MXene. We took a similar approach with investigating 1,4-phenylenediamine@$Ti_3C_2T_x$[69]. It is important to note that the PBE functional may affect the prediction of band alignment between extended materials and adsorbed molecules.

### The temperature differential spectra
The temperature differential spectra were obtained by subtracting a room-temperature absorption spectrum from the high-temperature steady-state absorption spectra. The detailed description can be found in Supplementary Note 2.

## Data availability
The authors declare that the data supporting the findings of this study are available within this published article and its supplementary. Source data are provided in this paper. Source data are provided with this paper.

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

## Acknowledgements

The experimental work was supported by the National Natural Science Foundation of China (Grant Nos. 22073003, 22303107, 22241304, 22225303), the National Natural Science Foundation of China (NSFC Center for Chemical Dynamics (Grant No. 22288201)), the Scientific Instrument Developing Project of the Chinese Academy of Sciences (Grant No. GJJSTD20220001), the Innovation Program for Quantum Science and Technology (2021ZD0303304), the Innovation Fund Project of Dalian Institute of Chemical Physics (DICP I202112). The Project was funded by China Postdoctoral Science Foundation (Grant No. 2023M732556, and the Fundamental Research Funds for the Central Universities.

## Author contributions

Q.Z., J.B.L. and K.J.Y. designed the experiments. Q.Z. performed the ultrafast experiment. W.L. and X.C. contributed to the theoretical calculations. Q.Z. and J.Z. performed the temperature differential spectra experiments. Q.Z., J.B.L., K.J.Y., R.X.Z., P.Z.T. and M.J.H. analyzed the data. G.R.W. and X.M.Y. discussed the results. All authors participated in the writing of the manuscript.

## Competing interests

The authors declare no competing interests.
