## [Peer Review File - NEW · Nature Communications]

Real-time observation of two distinctive non-thermalized hot electron dynamics at MXene/molecule interfacesReviewer #1 (Remarks to the Author):

Behavior of photoinjected electrons in MXenes and at MXene/molecule interfaces is an interesting and important topic that is relevant to many of the applications of MXenes in composite materials, photocatalysis and others.

Authors carried out an extensive set of controlled experiments, investigating transient optical absorption in aqueous solutions of T3C2 MXene alone and mixed with five different organic dyes with differing absorption profiles in the visible range. They find that photoexcitation of mixed solutions results in the onset of ground absorption bleach of dye molecules in the presence of MXene sheets, when excited with the photon energy that is unable to directly excite interband transitions in the dye molecules. This finding is a clear sign of molecule-MXene interactions and excitation transfer over ultrashort time scales.

Moreover, they find that the timescale of indirect excitation of dye molecules is ~ 2 times slower when the excitation photon energy is lower than the difference in energy between the MXene Fermi energy and the unoccupied MO, pointing at different mechanisms.

What I worry about is over-interpretation of (very interesting) data. The authors come up with four distinct excitation transfer pathways, as illustrated in Fig. 1. Those pathways are supposed to take place over different (but overlapping: 0-50 fs vs 0-125 fs) time scales. Those are plausible scenarios but the data presented in the paper is not sufficient to distinguish between them. A new proposed mechanism (non-thermalized electron induced direct heat transfer, that the authors claim involves scattering of a non-thermalized electron from a somewhat ill-defined MXene-molecule interface) is particularly difficult to follow and to distinguish from the non-thermalized electron transfer.

Figure 1 is more confusing than helpful, in my view. I find energy diagrams like those in Fig. 3 (c,f,i) more useful when trying to follow the story.

Can authors describe what they know or hypothesize about interaction between MXene sheets in solution with dye molecules, prior to photoexcitation? Is there covalent bonding? What role is the solvent playing? What MXene surface terminations were considered and what role do they play in MXene-molecule interaction? Schematic in Fig. 2a makes it seem like the electron and heat transfer occurs between the Ti atoms and molecules but Fig. 6 and DFT discussion alludes to formation of hybridized states. This aspect is critical to understanding the results presented here and the different proposed mechanisms but the discussion is essentially absent.

With this in mind, I would suggest to the authors to focus on clearly presenting their data, acknowledging what is and what isn't yet known, and possibly presenting several hypotheses regarding what they found.

Additional comments:

Re: MXene photoexcitation.

- Page 3, lines 76-77. "... while the heat transfer seems to be impossible" – What does that mean? Many papers on MXene photothermal properties and biomedical applications directly address heat transfer, and several have studied early-time dynamics (Guzelturk et al. *Angew. Chem. Int. Ed.* 2023, 23, 7, 2677–2686; Huang et al. *Nanophotonics*, 2020, 9, 2233-2249; and many others)

- Page 4, line 83: "relatively inert surface and narrowband plasmonic absorption" – MXene surface is not inert, as evidenced by its proposed catalytic applications. Plasmonic absorption in T13C2 is broadband, as authors point out themselves earlier ("visible to IR"). Moreover, studies have found that direct excitation of the plasmon resonance isn't needed to observe lattice heating and associated plasmon bleaching (Colin-Ulloa et al. 2023, *Adv. Mater.* 35 (8), 2208659)

My suggestion would be to significantly revise the manuscript before it can be published.

The manuscript also needs to be edited to improve clarity. Here are some specific comments:

- Abstract: "non-thermalized electron dynamical feature" – this is unclear, please specify what 'feature' is discussed.

- Page 3, line 78: remove 'a'

- Lots of typos, such as "800 nm exciation" (like 113 of SI), etc

-

Reviewer #2 (Remarks to the Author):

The manuscript by Qi Zhang et al. reports on ultrafast measurements on various molecules at the interface with a MXene. The results are rationalized in terms of different mechanisms, including non-thermalized electron transfer and non-thermalized electron scattering inducing heating of the molecule. The latter mechanism, in particular, was not experimentally characterized before.

I think the topic is timely and of broad interest, and the system well-chosen to gain insights into such complex phenomena. The experimental results are original, and, as far as I can judge, technically sound. As such, I think the work has the appropriate features to be published in Nature Communications. There are however different issues that should be properly addressed, chiefly related to the interpretation of the results :

1. In the whole paper, it is assumed that only hot electrons (as opposed to hot holes) can be transferred to the molecule, or can heat the molecule. Why hot holes are neglected?

2. The feature characterizing the TA of MXene, the broad positive band with peak at $\sim 550\text{nm}$ shown in Fig. 2d is assigned to the results of electron-electron scattering process leading to thermalized electrons, but no explanation is provided of what transitions originate the band itself. It is far from obvious, since typically plasmonic materials have TA featuring derivative shapes that originate from a shift of the plasmon peak position upon absorption of the pump. This is of course relevant for correctly interpreting the experiments

3. For the MB molecule -pump 800nm it is assumed that the change in the STA plot (fig. 3 b) at early times are due to non-thermalized electron injection. Leaving aside the point of electron vs hole injection raised above, why a resonant energy transfer (i.e., the energy from the excited plasmon is transferred to the neutral excitation in the molecule) is ruled out? The similarity of the 0-delay STA at 800nm (fig. 3b) and that assigned to the direct excitation of the molecule (fig. 3g) would suggest that.

4. The characterization of non-thermalized electron scattering inducing heating of the molecule (btw, this mechanism has been recently theoretically characterized in Vanzan et al. Nano Lett. 2023, 7, 2719) is certainly a prominent result of this work. As such, it requires to be properly backed up. In this respect, it would be important to have the ΔT absorption spectra of all the investigated molecules, not just R6G, to see if they match with the non-decaying component of the STA spectra. Moreover, the ΔT absorption spectra given in Sup. Fig. 10a has no indication of the sign of the ΔA (where is the zero on the y-axis?). I would expect both positive and negative features appear. Is this the case? If so, how does this compare with the all negative STA (Fig. 4b)? Would it be possible to measure how the absorption spectrum change with the temperature for the MXene/R6G system, rather than for R6G in solution, so to have a more direct comparison?

5. It is unclear if figures 6b and 6c provides projected DOS from a calculation of the entire MXene/molecule system or if the reported DOS refers to isolated MXene and molecule, respectively. In the former case, it is important to remind the reader the poor performance of PBE in predicting the band alignment between extended materials and adsorbed molecules; in the latter case, how was the alignment of the MXene and molecule DOS chosen?

6. p.11-12: the discussion of the mechanism of the energy transfer from non-thermalized electrons to molecular vibrations is quite confusing. "When the molecular nuclear motion resonates with the metal's energy level": which metal's energy level?. Refs 51-54 focus on the problem of energy transfer from molecular vibrations to the continuum of e-h pair excitations, here the energy flow is in the opposite direction. I think the mechanism described in Nano Lett. 2023, 7, 2719 should be also mentioned in this discussion

7. p.3: it is unclear to me what the Tx means in the definition of the MXene. Please clarify.

Reviewer #3 (Remarks to the Author):

The authors investigate electron/heat transfer at MXene/molecule interfaces. For this purpose, they employ time-resolved spectroscopy to simultaneously capture both MXene and molecules transient spectra. In general, analyzing the contribution of non-thermal and thermal electrons to electron/heat transfer is more difficult. This work clearly distinguishes the dynamic behavior of non-thermal and thermal electrons at the interfaces within 125 fs, providing a novel perspective to advance our current understanding of plasmon induced electron/heat transfer in 2D material interfaces. The data presented in this work are of good quality for obtaining nonthermal electron decay time and pathways. The analysis is mostly convincing, and thus the results are interesting, in principle, suitable for publication in Nature Communications. However, there are also some issues that the authors should address:

1. In this study, the authors observed two heat transfer channels (Pathway II and Pathway IV) in MXene/R6G. If both channels contribute to heating the molecule, why, after the optical excitation of MXene, can't MXene and R6G reach thermal equilibrium through Pathway IV? Why is R6G continually heated via Pathway II? Clarification from the authors is needed.
2. In Figure 3, observing the bleach signal of molecules may indicate electron transfer in the MXene/molecule system, but it could also result from various physical changes occurring around the molecule. The shift in molecule may arise from the Stark effect, wherein the excitation of surface plasmon induces the heating of charge carriers, leading to change in the dielectric environment of a molecule. This causes a shift in its maximum absorption peak. How do the authors eliminate the influence of the Stark effect in their study?
3. In Figure 3, the authors claimed that the measured molecular signal attributes to hot electron transfer. However, typically in a metal-molecule system, both hot electron and hot hole transfer are possible. How do the authors distinguish the observed molecular signal originating from hot electron transfer rather than hot hole transfer?
4. The authors show the different anisotropy behavior in MXene/MB and MXene/R6G. However, the control polarization-dependent TA experiments about MB and R6G are missing.
5. The authors show the SEM image MXene. However, the samples of MXene/molecule complexes are more important in this work. The SEM of MXene/molecule sample should also be shown.
6. The novel two channels (pathway III and pathway IV) were not observed in noble metal/molecule systems. The nature of the MXene/molecule interface fundamentally differs with the noble metal/molecule interface in both geometric and the electronic structures of the elements. Further discussion is necessary.
7. The authors mentioned that "it exhibits metallicity and plasmon features from the visible to the infrared band." Has this result been reported before? If not, please provide evidence.

Response to reviewer's comments

Reviewer #1 (Remarks to the Author):

Behavior of photoinjected electrons in MXenes and at MXene/molecule interfaces is an interesting and important topic that is relevant to many of the applications of MXenes in composite materials, photocatalysis and others. Authors carried out an extensive set of controlled experiments, investigating transient optical absorption in aqueous solutions of Ti_3C_2 MXene alone and mixed with five different organic dyes with differing absorption profiles in the visible range. They find that photoexcitation of mixed solutions results in the onset of ground absorption bleach of dye molecules in the presence of MXene sheets, when excited with the photon energy that is unable to directly excite interband transitions in the dye molecules. This finding is a clear sign of molecule-MXene interactions and excitation transfer over ultrashort time scales. Moreover, they find that the timescale of indirect excitation of dye molecules is ~ 2 times slower when the excitation photon energy is lower than the difference in energy between the MXene Fermi energy and the unoccupied MO, pointing at different mechanisms.

Q1. What I worry about is over-interpretation of (very interesting) data. The authors come up with four distinct excitation transfer pathways, as illustrated in Fig. 1. Those pathways are supposed to take place over different (but overlapping: 0-50 fs vs 0-125 fs) time scales. Those are plausible scenarios but the data presented in the paper is not sufficient to distinguish between them. A new proposed mechanism (non-thermalized electron induced direct heat transfer, that the authors claim involves scattering of a non-thermalized electron from a somewhat ill-defined MXene-molecule interface) is particularly difficult to follow and to distinguish from the non-thermalized electron transfer.

Author Reply: Thank you for your comments. We agree with you that the two ultrafast pathways (0-50 fs vs 0-125 fs) are not sufficient to distinguish between them from a time-resolved view. But the dynamics of electron and heat transfer are different, which could help us to distinguish the two pathways. The transferred electron would recombine within 1 ps leading to a fast decay (R Fig. 1 a), while the transferred heat would persist for an extended period (R Fig.1 b, a timescale of tens of picoseconds to nanoseconds) with non-decay in the initial 20 ps, according to previous reports^{1, 2}. In principle, the electrons transferred to the molecules will rapidly return to MXene. And then the electrons relax via the electron-phonon coupling in MXene. While the heat relaxes via the phonon-phonon interactions. Since the time of the electron-phonon coupling is significantly shorter than the phonon-phonon interactions, the dynamics of transferred electrons exhibit a rapid decay, whereas the dynamics of heat demonstrate a non-decay feature.

For our experimental result of STA-R6G excitation at 800 nm, the non-decay feature (Fig. 5c red circle data) is consistent with the transferred heat dynamics while mismatch with the transferred electron dynamics.

To refine the definition of non-thermalized electron-induced direct heat transfer, we modify the sentence “In contrast, the non-thermalized electrons engage in scattering at the interface, leading to the attached Rhodamine 6G (R6G) molecules heating within the initial 125 fs (Fig.1 **pathway IV**).” to “In contrast, in MXene/Rhodamine 6G (R6G) complex, the energy loss of non-thermalized electrons occurs via the electron scattering at MXene/R6G interface, resulting in heat transfer to attached R6G molecules within the initial 125 fs (Fig.1 **pathway II**).”

R Fig. 1. (a) TA kinetics of RhB GSB kinetics (empty circles) and the exponential fitting (blue line) in FICO–RhB complex. After infrared light excitation (red beam), the hot-electron transfers to the adsorbed RhB (red arrow). The data is from Nature Communications 2020, 11 (1), 2944. (b) Kinetics of the extracted RhB GSB of Cu_{2-x}Se -RhB (gray dots) and its fit (red line). The orange line is the scaled and shifted TA decay of Cu_{2-x}Se -RhB at 510 nm between -3 to 3 ps, representing the rise of the lattice temperature through the electron–phonon coupling of the NCs. The blue line is the scaled and inverted TA decay of Cu_{2-x}Se -RhB at 510 nm between 3 and 1300 ps, representing the heat transfer between the Cu_{2-x}Se NCs to the surrounding solvent. Schematic representation of the rapid heating of RhB on the surface of Cu_{2-x}Se -RhB complexes. The data is from Nano Letters 2021, 21 (1), 453–461. Panel a adapted with permission from Nature Communications 2020, 11 (1), 2944, and panel b adapted with permission from Nano Letters 2021, 21 (1), 453–461. Copyright American Chemical Society.

Q2. Figure 1 is more confusing than helpful, in my view. I find energy diagrams like those in Fig. 3 (c, f, i) more useful when trying to follow the story.

Author Reply: Thank you for your comments. To provide a clear description, we have revised Fig. 1. The conventional electronic/heat transfer pathway diagram (R Fig. 2) has been moved to the supplementary materials (Supplementary, Fig.1). The electronic/heat transfer pathways reported in this study (R Fig. 3) are now presented in Fig. 1 in the main text.

R Fig. 2(Supplementary Figure 1). The conventional electron and heat transfer pathways.

R Fig. 3 (Fig. 1). Schematic illustration presenting two different energy transfer dissipation pathways from the non-thermalized electron to the molecule at interface after optical excitation.

Q3. Can authors describe what they know or hypothesize about interaction between MXene sheets in solution with dye molecules, prior to photoexcitation? Is there covalent bonding? What role is the solvent playing?

Author Reply: Thank you for your comments. The ultrafast experiment results observed MXene/molecules film in the main text. We apologize for the unclear description of the method in the previous version.

For the solution situation, based on the previous reports^{3, 4, 5}, the MXene-dye molecules heterointerfaces in solution are mainly electrostatic interactions. In aqueous solution, the $Ti_3C_2T_x$ surfaces are negatively charged, which originates from the negatively charged surface terminations^{5, 6, 7}. For molecules, the aqueous solution makes these cation dye molecules dissociate, leading to molecules surface exhibiting a positive charge. The positively charged molecules and MXene with negatively charged surface would form the effective electrostatic interaction.

For MXene/molecule in aqueous solution, the solvent serves as the acceptor of the energy from the MXene/molecule after ~ 20 ps (R Fig. 4 a and b). There is the hydrogen bonding interaction between MXene and aqueous solution⁸, but it is weaker compared to the electrostatic interaction⁹. Therefore, the hydrogen bonding interaction between MXene and solution may be weaker slightly than the interaction between MXene and molecules.

R Fig. 4 MXene/R6G in aqueous solution after excitation at 800 nm. (a) STA-R6G spectra. (b) STA dynamics probed at 545 nm.

Q4. What MXene surface terminations were considered and what role do they play in MXene-molecule interaction? Schematic in Fig. 2a makes it seem like the electron and heat transfer occurs between the Ti atoms and molecules but Fig. 6 and DFT discussion alludes to formation of hybridized states.

Author Reply: Thank you for your comments. Different surface terminations of MXene were considered, including oxygen, hydroxyl, and fluorine. These terminations may play an important role in MXene-molecule interactions by influencing the surface properties of MXene, we performed calculations for $\text{Ti}_3\text{C}_2\text{F}_2$ /molecules. We have added the calculated result in Supplementary Figure 40. The results indicate that, compared to $\text{Ti}_3\text{C}_2\text{O}_2$ /molecules, there are some changes in charge distribution, adsorption energy, and interface distance. Hence, the surface terminations may change the electron/heat transfer rate and efficiency.

R Fig. 5 (Supplementary Figure 40) Calculated charge density differences of fluorine-terminated MXene/molecules (a) $\text{Ti}_3\text{C}_2\text{F}_2$ /MB. (b) $\text{Ti}_3\text{C}_2\text{F}_2$ /R6G.

We apologize for any confusion caused by the schematic diagram of Fig. 2a. The MXene in the figure is a simplified schematic diagram without surface terminations. The Figure aims to illustrate that the nonthermalized electrons on the MXene's surface could directly interact with molecules. We have changed the positions of electron-hole pairs in Fig. 2a (R Fig. 6), and provided a more detailed description: "The experiment for an ultrafast pump-probe approach uses a wavelength-switchable laser as the pump to generate nonthermal electron-hole pairs in MXene. Then a super-continuous white-light source is used to probe the molecular response at different time delay. Some electrons on MXene's surface may interact with molecules, leading to both electron and heat transfer. The remaining electrons decay in MXene, resulting in the lattice heating. The terminations on MXene surface are omitted in the simplified schematic diagram."

Thank you for your reminder. The description of the formation of hybridized states is inaccurate. We modify the sentence " In Fig. 6a, the charge density difference between MXene and R6G reveals a hybridized electronic orbital at interface " to "In Fig. 6a, the charge density difference reveals the redistribution of charges after the combination of R6G with MXene."

R Fig. 6 (Fig. 2a)

Q5. This aspect is critical to understanding the results presented here and the different proposed

mechanisms but the discussion is essentially absent. With this in mind, I would suggest to the authors to focus on clearly presenting their data, acknowledging what is and what isn't yet known, and possibly presenting several hypotheses regarding what they found.

Author Reply: We appreciate the reviewer's suggestion. Our results imply that different proposed mechanisms are dependent on three factors: the strength of interfacial interaction, the energy gap (ΔE) between MXene and the attached molecules, and the illuminating photon energy (E_{hv}). We have discussed the latter two factors in detail in the discussion section.

In the revised manuscript, we have added a detailed discussion about the first factor in the main text: "In addition, the strength of interfacial interaction between the MXene and the attached molecules is also crucial for both processes. Our DFT calculations (Fig. 6a and d) show the high adsorption energies (≥ 3.7 eV) and small distances (≤ 2.1 Å) between MXene and molecules, which indicate effective interfacial spatial wavefunction overlap. This overlap facilitates strong interfacial electron-electron and electron-nucleus interactions. Previous studies¹⁰ also reported that the MXene exhibits relatively strong interaction with molecules, including electrostatic, polymerization, and strong binding. The relatively strong interaction leads to **NET/NEIHT** mechanisms occurring at MXene/molecule interfaces. Furthermore, the structures of ultrathin layers also contribute to the occurrence of the **NET/NEIHT** pathways. In contrast, noble metals like silver and gold have fully occupied d orbitals, thus providing fewer electrons when interacting with molecules, leading to relatively weaker adsorptions with the adsorption energy being less than 1eV, and the distance between the molecule and the metal being greater than 2.9 Å¹¹. Besides, the ligand molecules on the surface of noble metal nanostructures further reduce the interaction with molecules. The weak interaction leads to the conventional electron/heat transfer pathways in most of noble metal/molecule systems."

Q6. Additional comments:

Re: MXene photoexcitation.

- Page 3, lines 76-77. "... while the heat transfer seems to be impossible" – What does that mean? Many papers on MXene photothermal properties and biomedical applications directly address heat transfer, and several have studied early-time dynamics (Guzelturk et al. Nano Lett. 2023, 23, 7, 2677–2686; Huang et al. Nanophotonics, 2020, 9, 2233-2249; and many others)

Author Reply: Thank you for your comments. We apologize for the unclear description of this sentence. We have revised this sentence to "...while completing ultrafast interfacial heat transfer in tens to hundreds of femtoseconds seems to be impossible."

Q7. - Page 4, line 83: "relatively inert surface an narrowband plasmonic absorption" – MXene surface is not inert, as evidences by its proposed catalytic applications. Plasmonic absorption in Ti_3C_2 is broadband, as authors point out themselves earlier ("visible to IR"). Moreover, studies have found that direct excitation of the plasmon resonance isn't needed to observe lattice heating and associated plasmon bleaching (Colin-Ulloa et al. 2023, Avd. Mater. 35 (8), 2208659)

Author Reply: Thank you for your comments. We apologize for our incorrect expression. We have revised this sentence to "It is also different from noble metal nanostructures in terms of geometry, relatively inert surface, and narrowband plasmonic absorption. The MXene advancements in atomically thin layers, the complex surface electronic properties^{10, 12, 13, 14, 15, 16} and broadband optical absorption characteristics¹⁷, which may provide..."

Q8. My suggestion would be to significantly revise the manuscript before it can be published.

Author Reply: Thank you for your comments. Based on your comments, we have performed new experiments and calculations to improve our manuscript.

Q9. The manuscript also needs to be edited to improve clarity.

Here are some specific comments:

- Abstract: “non-thermalized electron dynamical feature” – this is unclear, please specify what ‘feature’ is discussed.
- Page 3, line 78: remove ‘a’
- Lots of typos, such as “800 nm exciation” (like 113 of SI), etc

Author Reply: Thank you for your comments. The non-thermalized electron dynamical feature includes interfacial non-thermalized electron transfer and heat transfer before electron thermalization. The two processes exhibited different dynamics features: fast decay and non-decay dynamics signal within 1 ps. Maybe using the word “feature” may not be appropriate. We have removed the word "feature"

Thank you for your careful review. We have corrected these errors.

Reviewer #2 (Remarks to the Author):

The manuscript by Qi Zhang et al. reports on ultrafast measurements on various molecules at the interface with a MXene. The results are rationalized in terms of different mechanisms, including non-thermalized electron transfer and non-thermalized electron scattering inducing heating of the molecule. The latter mechanism, in particular, was not experimentally characterized before.

I think the topic is timely and of broad interest, and the system well-chosen to gain insights into such complex phenomena. The experimental results are original, and, as far as I can judge, technically sound. As such, I think the work has the appropriate features to be published in Nature Communications. There are however different issues that should be properly addressed, chiefly related to the interpretation of the results:

Q1. In the whole paper, it is assumed that only hot electrons (as opposed to hot holes) can be transferred to the molecule, or can heat the molecule. Why hot holes are neglected?

Author Reply: Thank you for your comments. We added the discussion about the hot hole transfer in Supplementary Note 4: “The results suggest that the hot electrons can transfer to the molecules, or can heat the molecules at the MXene/molecules interface in this study. While the hot holes are neglected based on two reasons: (1) In general, for metallic material, the main carrier is electron since the electron has higher mobility. The faster relaxation dynamics and lower mobility of hot holes make them harder to transfer when compared with hot electrons¹⁸; (2) As shown in Supplementary Fig. 26, the hot hole generated by a pump pulse would first transfer to the HOMO+2 state. Our TA spectra could probe the bleaching signal from the response between HOMO and LUMO. The response of other transitions (such as the response of transitions from HOMO+2 to LUMO, ~2.5 eV) is not observed within the detection range (470-740 nm, 1.7-2.6 eV) of our experiment.

R Fig. 7 (Supplementary Figure 26). Partial density of states (TDOS) of entire MXene/MB. The red arrows represent a photo-induced generation of a hot hole, and the orange arrows indicate the probing wavelength.

Q2. The feature characterizing the TA of MXene, the broad positive band with peak at ~550nm shown in Fig. 2d is assigned to the results of electron-electron scattering process leading to thermalized electrons, but no explanation is provided of what transitions originate the band itself. It is far from obvious, since typically plasmonic materials have TA featuring derivative shapes that originate from a shift of the plasmon peak position upon absorption of the pump. This is of course relevant for correctly interpreting the experiments.

Author Reply: Thank you for your comments. We agree that “typically plasmonic materials have TA featuring derivative shapes that originate from a shift of the plasmon peak position upon absorption of the pump.” But photo-induced inter-band transition enhancement may also play a key factor in MXene. “Based on our DFT calculated electron structures (R Fig. 8a) and previous reports about optical properties of MXene¹⁹, the inter-band transient absorption covers the range from visible to near-infrared. The green and red arrows (R Fig 8 a) represent electronic inter-band transitions after absorbing visible and near-infrared photons, respectively.

We added this discussion in the main text “The broad positive band of 500-550 nm originates inter-band transition according to the calculated band structures of MXene (Supplementary Fig. 5a). The electron-electron scattering or heat make an increased number of electrons near Fermi level, resulting in the enhanced inter-band transitions as shown in Fig. 2d. The photo-induced absorption peak of ~510 nm for MXene in TA spectra derives from the overlap between inter-band absorption and plasmonic band (Supplementary Fig. 5b).”

R Fig. 8 (Supplementary Figure 5). (a) The calculated band structures of $\text{Ti}_3\text{C}_2\text{O}_2$ MXene. The green arrow represents the possible inter-band transitions of electrons in MXene after absorption of visible light. While the red arrow represents near-infrared light absorption. (b) A steady-state absorption spectrum of MXene.

Q3. For the MB molecule -pump 800nm it is assumed that the change in the STA plot (fig. 3 b) at early times are due to non-thermalized electron injection. Leaving aside the point of electron vs hole injection raised above, why a resonant energy transfer (i.e., the energy from the excited plasmon is transferred to the neutral excitation in the molecule) is ruled out? The similarity of the 0-delay STA at 800nm (fig. 3b) and that assigned to the direct excitation of the molecule (fig. 3g) would suggest that.

Author Reply: Thank you for your comments. The polarized-dependent TA experiment of MXene/MB with excitation at 745 nm suggests that direct excitation of the molecule is dominated. The direct excitation of molecules by photons is an instantaneous process (0 fs). While the plasmonic resonance energy transfer should be an ultrafast process of ~ 10 fs via plasmon decay. The time resolution of our ultrafast spectroscopy is not sufficient to distinguish the two processes. However, the polarization dependent responses of these two processes are different. The Polarization-dependent experiments can distinguish them.

We added this part in Supplementary Note 5: “The plasmon resonant energy transfer (i.e., the energy from the excited plasmon is transferred to the neutral excitation in the molecule) is ruled out based on the polarization-dependent experiment. The polarized dependent TA experiment of MXene/MB shows almost isotropic for the MXene-contributed signals (500-600 nm in Supplementary Fig. 27a) while strong anisotropic for the STA-MB contributed signal (650-740 nm in Supplementary Fig. 27b). And the polarized dependent TA of MB (Supplementary Fig. 28a-c) show significantly anisotropic. If the STA-MB signals come from plasmon resonant energy transfer, the polarized dependent STA-MB signal should exhibit isotropic due to the isotropic signals of MXene. This suggests that the direct excitation of the molecule is dominated.

R Fig. 9 (Supplementary Figure 27). Polarization-dependent TA and STA spectra in MXene/MB with excitation at 745 nm. (a) P- and S-polarized TA at 0 fs. (b) P- and S-polarized STA at 0 fs. The selected excitation at 745 nm rather than at 700 nm was caused by the strong noise of pump scattering, which hindered us from obtaining accurate data ranging from 680-720 nm.

Q4. The characterization of non-thermalized electron scattering inducing heating of the molecule (btw, this mechanism has been recently theoretically characterized in Vanzan et al. Nano Lett. 2023, 7, 2719) is certainly a prominent result of this work. As such, it requires to be properly backed up. In this respect, it would be important to have the DeltaT absorption spectra of all the investigated molecules, not just R6G, to see if they match with the non-decaying component of the STA spectra. Moreover, the DeltaT absorption spectra given in Sup. Fig. 10a has no indication of the sign of the Delta A (where is the zero on the y-axis?). I would expect both positive and negative features appear. Is this the case? If so, how does this compare with the all negative STA (Fig. 4b)? Would it be possible to measure how the absorption spectrum change with the temperature for the MXene/R6G system, rather than for R6G in solution, so to have a more direct comparison?

Author Reply: We thank you for pointing out the missing references. The theoretical research by Vanzan et al. has significantly inspired us. We have included a discussion related to this paper in the main text as follows: **“Furthermore, the mechanism of non-thermalized electron scattering inducing heating of the molecule revealed in this work, aligns with the pioneer's theoretical predictions²⁰, in which the energy of non-thermalized hot electrons could be released into molecular nuclear vibrations.”**

We show the zero on the y-axis of DeltaT absorption spectra of MXene/R6G in aqueous solution (R Fig. 10a and b).

R Fig. 10 (Supplementary Figure 13). Comparison between a temperature differential spectrum and a STA spectrum of MXene/R6G in aqueous solution. (a) Temperature differential spectrum (obtained by subtracting a steady-state absorption spectrum at high temperature from one at low temperature). **(b)**The STA-R6G spectrum at 150 fs after excitation at 800 nm.

You are right that the ΔT absorption spectra exhibit both positive and negative features appearing at high-temperature differences ($\Delta T > 40$ K), and only negative features appear at low-temperature differences ($\Delta T < 40$ K). We have measured the ΔT absorption spectra of MXene/R6G film, MXene/R6G in solution, R6G film, and R6G in solution. The results are shown in R Fig 11-13. The ΔT absorption spectra of MXene/R6G and R6G are different. The ΔT absorption spectra of MXene/R6G show a negative peak at ~ 545 nm (R Fig. 11 and R Fig. 12 a-c), while R6G exhibits a positive peak at ~ 545 nm (R Fig. 13b-c).

We added this part in the Supplementary Note 2: “To demonstrate universality, the observation of STA spectra may be caused by heated molecules, we have performed the differential absorption spectra of all the investigated MXene/molecules as shown in Supplementary Figs. 24 and 25. The differential absorption spectra at low-temperature differences ($\Delta T < 40$ K) exhibit only negative peaks. While they show both positive and negative peaks at high-temperature differences ($\Delta T > 40$ K). The results are almost consistent with STA-molecules spectra (Supplementary Figs. 20-22).”

R Fig 11. The differential absorption spectra of MXene/R6G film. The lower temperature difference data ΔT is less than 40 K; The higher temperature difference data ΔT is larger than 40 K.

R Fig. 12 (Supplementary Figure 25). MXene/R6G in aqueous solution. (a-c) Various temperature differential ($\Delta T=15, 25, 45$ K) absorption spectra (d) STA-R6G spectra. (e) STA-R6G dynamics probed at 545 nm.

R Fig. 13. (a) The Steady absorption spectrum of R6G in aqueous solution. (b) The differential absorption spectra of R6G in aqueous solution. (c) The differential absorption spectra of R6G film.

R Fig. 14 (Supplementary Figure 24). High- and low-temperature difference ($\Delta T < 40$ K and $\Delta T > 40$ K) spectra of MXene/molecules film. (a&b) MXene/R6G film. (c&d) MXene/RhB film. (e&f) MXene/SR101 film. (g&h) MXene/CV film. (i&j) MXene/MB film. These pure molecular temperature differential absorption spectra in MXene/molecules are obtained by subtracting the temperature differential absorption spectra of MXene/molecule (Supplementary Note 2).

Q5. It is unclear if figures 6b and 6c provides projected DOS from a calculation of the entire MXene/molecule system or if the reported DOS refers to isolated MXene and molecule, respectively. In the former case, it is important to remind the reader the poor performance of PBE in predicting the band alignment between extended materials and adsorbed molecules; in the latter case, how was the alignment of the MXene and molecule DOS chosen?

Author Reply: Thank you for your comments. To correct the confusing description, we have revised the sentence “the DOS for R6G, presented in Fig 6b...” to “**In the entire MXene/R6G system, the DOS for R6G, presented in Fig 6b...**”.

We calculate the DOS of the entire MXene/molecule system. As pointed out by the reviewer, it is well known that the PBE functional fails to accurately describe the non-local and strongly correlated effects of electron interactions. The PBE tends to underestimate the energy gap of the material and performs poorly on band alignment predictions. Nevertheless, this method was used in other literature (Chem. Mater. 2020, 32, 7884–7894 and Adv. Electron. Mater. 2021, 7, 2001202) for analyzing similar systems. A more precise method requires much greater efforts and resource that goes beyond the scope of this work. In main text, we add sentences in the DFT Calculation Methods section: “**We took a similar approach for investigating 1,4-phenylenediamine@Ti₃C₂T_x²¹. It is important to note that the PBE functional may affect the prediction of band alignment between the extended materials and adsorbed molecules.**”

Q6. p.11-12: the discussion of the mechanism of the energy transfer from non-thermalized electrons to molecular vibrations is quite confusing. “When the molecular nuclear motion resonates with the metal's energy level”: which metal's energy level? Refs 51-54 focus on the problem of energy transfer from molecular vibrations to the continuum of e-h pair excitations, here the energy flow is in the opposite direction. I think the mechanism described in Nano Lett. 2023, 7, 2719 should be also mentioned in this discussion

Author Reply: We thank you for pointing out the missing references. The theoretical research by Vanzan et al. has significantly inspired us. The energy levels are metal's electronic transitions. We have rewritten the sentence as follows: “**When the molecular nuclear motion resonates with the energy levels of the metal's electronic transitions...**”

We have included a discussion related to this paper in the main text as follows: “**Previous study results show the energy transfer from molecular vibrations to the continuum of e-h pair excitations^{22, 23, 24}. While our results exhibited that the energy flow is in the opposite direction.**” “**Furthermore, the mechanism of non-thermalized electron scattering inducing heating of the molecule revealed in this work, aligns with the pioneer's theoretical predictions²⁰, in which the energy of non-thermalized hot electrons could be released into molecular nuclear vibrations.**”

Q7. p.3: it is unclear to me what the Tx means in the definition of the MXene. Please clarify.

Author Reply: We have included the definition of Tx in the main text. “**T_x represents the surface terminated group of MXene, such as -O, -F, -OH.**”

Reviewer #3 (Remarks to the Author):

The authors investigate electron/heat transfer at MXene/molecule interfaces. For this purpose, they employ time-resolved spectroscopy to simultaneously capture both MXene and molecules transient

spectra. In general, analyzing the contribution of non-thermal and thermal electrons to electron/heat transfer is more difficult. This work clearly distinguishes the dynamic behavior of non-thermal and thermal electrons at the interfaces within 125 fs, providing a novel perspective to advance our current understanding of plasmon induced electron/heat transfer in 2D material interfaces. The data presented in this work are of good quality for obtaining nonthermal electron decay time and pathways. The analysis is mostly convincing, and thus the results are interesting, in principle, suitable for publication in Nature Communications. However, there are also some issues that the authors should address:

Author Reply: We appreciate the reviewer for the positive comments about this topic. We have performed substantial new experiments and analyses to improve our manuscript as shown below.

Q1. In this study, the authors observed two heat transfer channels (Pathway II and Pathway IV) in MXene/R6G. If both channels contribute to heating the molecule, why, after the optical excitation of MXene, can't MXene and R6G reach thermal equilibrium through Pathway IV? Why is R6G continually heated via Pathway II? Clarification from the authors is needed.

Author Reply: Thank you for your comments. Based on the calculated area ratio, only 10-15 % of the molecules cover the MXene surface. This means that as long as 10-15 % of the absorbed photon energy can be transferred to R6G through the NEIHT pathway. A significant amount of heat (85-90%) still exists in MXene that has not been covered by molecules. Therefore, the heat in this portion of MXene would be transferred to the molecules through the PMHT pathway.

We are adding the discussion about the question in Supplementary Note 6: **“The R6G is continually heated via the PMHT because the molecules cover only a small portion of the MXene surface (Supplementary Note 1), and only a portion of the absorbed photon energy can be transferred to R6G through the NEIHT pathway. While most of the heat still exists in MXene that has not been covered by molecules. Therefore, the heat in this portion of MXene would be transferred to the molecules through the PMHT pathway.”**

Q2. In Figure 3, observing the bleach signal of molecules may indicate electron transfer in the MXene/molecule system, but it could also result from various physical changes occurring around the molecule. The shift in molecule may arise from the Stark effect, wherein the excitation of surface plasmon induces the heating of charge carriers, leading to change in the dielectric environment of a molecule. This causes a shift in its maximum absorption peak. How do the authors eliminate the influence of the Stark effect in their study?

Author Reply: Thank you for your comments. In principle, a shift in its maximum absorption peak induced by the optical Stark effect would show a feature of derivative shapes in transient spectra^{25, 26}. While the STA spectra exhibit the feature of bleaching signals within 125 fs rather than the derivative shapes. Hence, we could rule out the influence of the Stark effect. In general, the optical Stark effect occurs in electronic excited semiconductor nanostructure^{25, 26} owing to the strong Coulomb field of electron-hole pairs. However, the abundant free electrons in metallic MXene would screen the Coulomb interaction of electron-hole pairs.

Q3. In Figure 3, the authors claimed that the measured molecular signal was attributes to hot electron transfer. However, typically in a metal-molecule system, both hot electron and hot hole transfer are possible. How do the authors distinguish the observed molecular signal originating from hot electron

transfer rather than hot hole transfer?

Author Reply: Thank you for your comments. The reply is as same as that in the **Author's Reply to Reviewer 2, Q1**.

Q4. The authors show the different anisotropy behavior in MXene/MB and MXene/R6G". However, the control polarization-dependent TA experiments about MB and R6G are missing.

Author Reply: Thank you for your comments. We have added the polarization-dependent TA experiments results of MB and R6G in Supplementary Figure 28. The results indicate that these molecules exhibit clear anisotropy.

R Fig. 15 (Supplementary Figure 28). Polarization-dependent transient spectra and dynamics of MB and R6G in aqueous solution and R6G film. (a-c) MB in aqueous solution after 620 nm excitation, P- and S-polarized dynamics probed at 670 nm. (d-f) R6G in aqueous solution after 510 nm excitation, P- and S-polarized dynamics probed at 530 nm. (g-i) R6G film with excitation at 510 nm, P- and S-polarized dynamics probed at 530 nm.

Q5. The authors show the SEM image MXene. However, the samples of MXene/molecule complexes are more important in this work. The SEM of MXene/molecule sample should also be shown.

Author Reply: Thank you for your comments. We have added SEMs of MXene/R6G and MXene/MB in Supplementary Figure 3.

R Fig. 16 (Supplementary Figure 3). (a) TEM image of MXene/R6G. (b) Energy dispersive spectroscopy (EDS) result of the MXene/R6G sample. The element of Cl comes from R6G. (c) TEM image of MXene/MB.

Q6. The novel two channels (pathway III and pathway IV) were not observed in noble metal/molecule systems. The nature of the MXene/molecule interface fundamentally differs with the noble metal/molecule interface in both geometric and the electronic structures of the elements. Further discussion is necessary.

Author Reply: Thank you for your comments. The discussion of different mechanisms is as follows: “It is also different from noble metal nanostructures in terms of geometry, relatively inert surface, and narrowband plasmonic absorption. The metallic MXene advancements in atomically thin layers, the complex surface electronic properties^{10, 12, 13, 14, 15, 16} and broadband optical absorption characteristics¹⁷...” Further discussion can be found in **the Author's Reply of Reviewer 1, Q5**.

Q7. The authors mentioned that “it exhibits metallicity and plasmon features from the visible to the infrared band.” Has this result been reported before? If not, please provide evidence.

Author Reply: Thank you for your comments. The metallicity and plasmon features have been well-studied previously. We have cited the reference from *ACS Nano* **12**, 8485-8493 (2018).

References

1. Zhou D, Li X, Zhou Q, Zhu H. Infrared driven hot electron generation and transfer from non-noble metal plasmonic nanocrystals. *Nat. Commun.* **11**, 2944 (2020).
2. Yang W, Liu Y, McBride JR, Lian T. Ultrafast and long-lived transient heating of surface adsorbates on plasmonic semiconductor nanocrystals. *Nano Lett.* **21**, 453-461 (2021).
3. Lim S, *et al.* Role of electrostatic interactions in the adsorption of dye molecules by Ti₃C₂-MXenes. *RSC Adv.* **11**, 6201-6211 (2021).
4. Sarycheva A, *et al.* Two-dimensional titanium carbide (MXene) as surface-enhanced Raman scattering substrate. *J. Phys. Chem. C* **121**, 19983-19988 (2017).
5. Ren CE, Hatzell KB, Alhabeb M, Ling Z, Mahmoud KA, Gogotsi Y. Charge- and Size-Selective Ion Sieving Through Ti₃C₂T_x MXene membranes. *J. Phys. Chem. Lett.* **6**, 4026-4031 (2015).
6. Riazi H, *et al.* Surface modification of a MXene by an aminosilane coupling agent. *Adv. Mater. Interfaces* **7**, 1902008 (2020).
7. Zaman W, *et al.* In situ investigation of water on MXene interfaces. *Proc. Natl. Acad. Sci. U. S. A.* **118**, e2108325118 (2021).
8. Zhang Q, *et al.* Surface Oxidation Modulates the interfacial and lateral thermal migration of MXene (Ti₃C₂T_x) Flakes. *J. Phys. Chem. Lett.* **11**, 9521-9527 (2020).

9. Li J, *et al.* Ultrafast flash energy conductance at MXene-surfactant interface and its molecular origins. *Adv. Mater. Interfaces* **6**, 1901461 (2019).
10. Boota M, *et al.* Mechanistic understanding of the interactions and pseudocapacitance of multi-electron redox organic molecules sandwiched between MXene layers. *Adv. Electron. Mater.* **7**, 2001202 (2021).
11. Liu W, Tkatchenko A, Scheffler M. Modeling adsorption and reactions of organic molecules at metal surfaces. *Acc. Chem. Res.* **47**, 3369-3377 (2014).
12. Naguib M, *et al.* Two-dimensional nanocrystals produced by exfoliation of Ti_3AlC_2 . *Adv. Mater.* **23**, 4248-4253 (2011).
13. Zheng W, *et al.* Band transport by large Fröhlich polarons in MXenes. *Nat. Phys.* **18**, 544-550 (2022).
14. Guzelturk B, *et al.* Understanding and controlling photothermal responses in MXenes. *Nano Lett.* **23**, 2677-2686 (2023).
15. Velusamy DB, *et al.* MXenes for plasmonic photodetection. *Adv. Mater.* **31**, 1807658 (2019).
16. Colin-Ulloa E, *et al.* Ultrafast spectroscopy of plasmons and free carriers in 2D MXenes. *Adv. Mater.* **35**, 2208659 (2023).
17. Zhang Q, *et al.* Simultaneous capturing phonon and electron dynamics in MXenes. *Nat. Commun.* **13**, 7900 (2022).
18. Zhang Y, Guo W, Zhang Y, Wei WD. Plasmonic photoelectrochemistry: in view of hot carriers. *Adv. Mater.* **33**, 2006654 (2021).
19. Berdiyrov GR. Optical properties of functionalized $Ti_3C_2T_2$ (T = F, O, OH) MXene: first-principles calculations. *AIP Adv.* **6**, 055105 (2016).
20. Vanzan M, Gil G, Castaldo D, Nordlander P, Corni S. Energy transfer to molecular adsorbates by transient hot electron spillover. *Nano Lett.* **23**, 2719-2725 (2023).
21. Boota M, *et al.* Probing molecular interactions at MXene-organic heterointerfaces. *Chem. Mater.* **32**, 7884-7894 (2020).
22. Rittmeyer SP, Meyer J, Reuter K. Nonadiabatic vibrational damping of molecular adsorbates: insights into electronic friction and the role of electronic coherence. *Phys. Rev. Lett.* **119**, 176808 (2017).
23. Kumar S, Jiang H, Schwarzer M, Kandratsenka A, Schwarzer D, Wodtke AM. Vibrational relaxation lifetime of a physisorbed molecule at a metal surface. *Phys. Rev. Lett.* **123**, 156101 (2019).
24. Li J, *et al.* Two distinctive energy migration pathways of monolayer molecules on metal nanoparticle surfaces. *Nat. Commun.* **7**, 10749 (2016).
25. Yong C-K, *et al.* Biexcitonic optical Stark effects in monolayer molybdenum diselenide. *Nat. Phys.* **14**, 1092-1096 (2018).
26. Li Y, He S, Luo X, Lu X, Wu K. Strong spin-selective optical stark effect in lead halide perovskite quantum dots. *J. Phys. Chem. Lett.* **11**, 3594-3600 (2020).

Reviewer #1 (Remarks to the Author):

The authors addressed all my concerns and answered all questions. As it stands now, the paper is well-organized and presents exciting new results.

Reviewer #2 (Remarks to the Author):

I went through the rebuttal and the revised version of the manuscript. While most of my comments have been successfully addressed, there are still open issues that I think should be clarified:

1. I do not find the case for excluding hot-holes role vs hot-electrons compelling. The reference cited in the section of the SI (p.25, ref. 6) is actually showing how widespread is the role of hot holes. The discussion based on the plot of the density of states (SI fig. 26) is not so useful. First, it assumes that a) DOS calculated by PBE xc functional are accurate enough for setting properly the molecular levels vs the MXene features and b) that a single particle picture that neglect plasmonic effects can be applied. Both these assumptions are rather bold. Second, even taking these assumptions as valid, the transition shown in that picture is not necessarily the main source of holes: it is the joint density of states, not only the DOS of holes, to determine at what energies holes and electrons form. As such, I think the possible role of hot holes is still unclear. Is it really pivotal for the main conclusion of this work (i.e., the likeliness of a mechanism where energy is transferred to molecular vibrations by hot carriers) to establish definitely if the relevant hot carriers are hot electrons instead of hot holes? I think the paper would keep its impact even leaving this issue not fully settled

2. To rule out the excitation of MB by a resonant energy transfer (RET) mechanism from the excited plasmon, reference is made to the polarized TA measurements. If I understood correctly (if not, please clarify, also in the text), the justification is: the TA of MXene is isotropic, the TA of MB is anisotropic, then if the origin of the STA of the MB contribution would be resonant energy transfer, such STA should be isotropic as well, which is not. However, I do not see this as a strong argument: resonant energy transfer may happen from the excited MXene plasmon before the damping processes that make the subsequent TA signal isotropic, conserving the anisotropic behavior for the MB TA.

In short, I think there are still points that are given as conclusive while they are not. My recommendation, in line with that of another reviewer, is to avoid overinterpretation, presenting explanations as perhaps likely hypothesis but not as certain when there is no conclusive evidence. I also urge the authors to improve the English throughout the manuscript, in different places it is difficult to grasp the meaning of sentences.

Reviewer #3 (Remarks to the Author):

The authors have addressed all my questions. I suggest it can be published.

Response to reviewer's comments

Reviewer #1 (Remarks to the Author):

The authors addressed all my concerns and answered all questions. As it stands now, the paper is well-organized and presents exciting new results.

Author Reply: Thank you for your thorough review.

Reviewer #2 (Remarks to the Author):

I went through the rebuttal and the revised version of the manuscript. While most of my comments have been successfully addressed, there are still open issues that I think should be clarified:

1. I do not find the case for excluding hot-holes role vs hot-electrons compelling. The reference cited in the section of the SI (p.25, ref. 6) is actually showing how widespread is the role of hot holes. The discussion based on the plot of the density of states (SI fig. 26) is not so useful. First, it assumes that a) DOS calculated by PBE xc functional are accurate enough for setting properly the molecular levels vs the MXene features and b) that a single particle picture that neglect plasmonic effects can be applied. Both these assumptions are rather bold. Second, even taking these assumptions as valid, the transition shown in that picture is not necessarily the main source of holes: it is the joint density of states, not only the DOS of holes, to determine at what energies holes and electrons form. As such, I think the possible role of hot holes is still unclear. Is it really pivotal for the main conclusion of this work (i.e., the likeliness of a mechanism where energy is transferred to molecular vibrations by hot carriers) to establish definitely if the relevant hot carriers are hot electrons instead of hot holes? I think the paper would keep its impact even leaving this issue not fully settled.

In short, I think there are still points that are given as conclusive while they are not. My recommendation, in line with that of another reviewer, is to avoid overinterpretation, presenting explanations as perhaps likely hypothesis but not as certain when there is no conclusive evidence. I also urge the authors to improve the English throughout the manuscript, in different places it is difficult to grasp the meaning of sentences.

Author Reply: Thank you for your suggestion. We agree with your viewpoint. In the revised manuscript, we have added a detailed discussion about the roles of hot holes in the main text: “Additionally, nonthermalized hot holes may also inject into molecules or transfer energy to molecular vibrations. In any case, our systematic studies reveal that the energy flow pathway from non-thermalized carrier to attached molecules”. Besides, we modify the discussion about the hot hole transfer in Supplementary Note 4. We changed the sentence of “...While hot holes are neglected. This consideration was based on two reasons:...” to “We suppose that the transfer of hot holes may be more difficult than the transfer of hot electrons. This hypothesis was based on two reasons...”. And we added a sentence “But, at present, there is no certain evidence. The role of hot hole transfer is still further explored.”

2. To rule out the excitation of MB by a resonant energy transfer (RET) mechanism from the excited plasmon, reference is made to the polarized TA measurements. If I understood correctly (if not, please clarify, also in the text), the justification is: the TA of MXene is isotropic, the TA of MB is anisotropic, then if the origin of the STA of the MB contribution would be resonant energy transfer, such STA should be isotropic as well, which is not. However, I do not see this as a strong argument: resonant energy transfer may happen from the excited MXene plasmon before the damping

processes that make the subsequent TA signal isotropic, conserving the anisotropic behavior for the MB TA.

Author Reply: Thank you for your comments. Your understanding is correct. We agree with your point that the argument from polarized TA results is not sufficiently strong. We have revised the sentence in the main text "..., representing the direct excitation of MB molecules, as shown in Fig. 3i." to "..., representing the direct excitation of MB molecules (Fig. 3i) or resonant energy transfer.". Additionally, we modify the discussion about the hot hole transfer in Supplementary Note 5: The sentence of "The plasmon resonant energy transfer (i.e., the energy from the excited plasmon is transferred to the neutral excitation in the molecule) **is ruled out** based on the polarization-dependent experiment." was changed to "**We discuss the plasmon resonant energy transfer (i.e., the energy from the excited plasmon is transferred to the neutral excitation in the molecule) about the results of Fig. 3 g & h based on the polarization-dependent experiment.**". We modify the sentence of "This suggests that the direct excitation of the molecule is dominated." to "**This result may indicate that the resonant energy transfer is not dominated.**"

Reviewer #3 (Remarks to the Author):

The authors have addressed all my questions. I suggest it can be published.

Author Reply: Thank you for your assessment.